# Ubiquity and impact of thin mid-level clouds in the tropics

Quentin Bourgeois[1,2], Annica M.L. Ekman[1,2], Matthew R. Igel[3] & Radovan Krejci[2,4]

Clouds are crucial for Earth's climate and radiation budget. Great attention has been paid to low, high and vertically thick tropospheric clouds such as stratus, cirrus and deep convective clouds. However, much less is known about tropospheric mid-level clouds as these clouds are challenging to observe *in situ* and difficult to detect by remote sensing techniques. Here we use Cloud-Aerosol Lidar with Orthogonal Polarization (CALIOP) satellite observations to show that thin mid-level clouds (TMLCs) are ubiquitous in the tropics. Supported by high-resolution regional model simulations, we find that TMLCs are formed by detrainment from convective clouds near the zero-degree isotherm. Calculations using a radiative transfer model indicate that tropical TMLCs have a cooling effect on climate that could be as large in magnitude as the warming effect of cirrus. We conclude that more effort has to be made to understand TMLCs, as their influence on cloud feedbacks, heat and moisture transport, and climate sensitivity could be substantial.

[1] Department of Meteorology, Stockholm University, Stockholm 10691, Sweden. [2] Bolin Centre for Climate Research, Stockholm University, Stockholm 10691, Sweden. [3] Department of Atmospheric Sciences, University of Miami, Miami, Florida 33149, USA. [4] Department of Applied Environmental Science and Analytical Chemistry, Stockholm University, Stockholm 10691, Sweden. Correspondence and requests for materials should be addressed to Q.B. (email: quentin.bourgeois@misu.su.se).

The radiative balance of the Earth is strongly influenced by clouds[1]. In general, clouds with a large optical thickness reflect most of the incoming shortwave radiation inducing a cooling effect, while clouds with a low cloud-top temperature trap outgoing long-wave radiation inducing a warming effect[2,3]. As a consequence, moderately thick low-level clouds (for example, stratocumulus) have a negative radiative effect, while thin high-level clouds (for example, cirrus) have a positive radiative effect[2]. As a long-term and global average, clouds cool the climate[1]. Under a perturbation of the radiative balance of the Earth, for instance by an increase in greenhouse gas concentrations, it is crucial to know how clouds would respond. However, representing clouds in global climate models is a challenge, and some models show that clouds would enhance the temperature response while others show that clouds would dampen the temperature change[1,4]. It is therefore essential to better understand cloud properties and formation processes, in particular in convective regions as highlighted in the last IPCC report[1].

The tropical convective cloud distribution can be characterized by a trimodal structure composed of shallow cumulus, congestus and cumulonimbus[5]. However, it has also been suggested that mid-tropospheric detrainment in the tropics at the freezing level (that is, at the 0 °C isotherm)[6,7] could form thin mid-level clouds (TMLCs). Using remote sensing techniques, TMLCs have been observed over Darwin (Australia)[8] and over West Africa[9]. In these areas, the TMLCs were found to be thinner than 2 km and to consist mainly of supercooled liquid water with a cloud-top height between 4 and 8 km and cloud-top temperatures ranging between 0 and -12.5 °C (refs 9,10). High-resolution regional modelling simulations over the tropical western Pacific have also indicated that TMLCs might appear frequently in the maritime tropics[11]. Nevertheless, despite these few individual observations of TMLCs, there is no general picture of their overall prevalence and their potential effect on climate has not yet been quantified.

Here we present a comprehensive study characterizing the physical and optical properties of TMLCs in the tropics using extinction values retrieved from the Cloud-Aerosol Lidar with Orthogonal Polarization (CALIOP) instrument[12] (see Methods for details) over a period of 5 years (2008–2012). A high-resolution model is used to examine their formation and a cloud radiative forcing model is used to evaluate their radiative effect. We find that TMLCs are formed across the tropics by detrainment during deep convection and that the clouds generally cool the climate.

## Results

**Physical and optical properties of TMLCs.** TMLCs were defined as clouds between 3 and 8 km and were assumed to not extend below or above these altitudes, respectively. A typical example of TMLCs is shown in Fig. 1. In this example, CALIOP indicates that TMLCs are between a few hundred metres and 2 km thick and occur mostly between 5 and 7 km altitude. In agreement with CALIOP data, the MODIS satellite[13] reports clouds with a top temperature of about 270 K. Distributions of observed morphological properties of TMLCs are shown in Fig. 2 and summarized in Table 1. Overall, TMLCs are thicker and higher during night than during day, and they are mostly liquid (86% according to the CALIOP algorithm of classification[14]). Their thickness ranges from 400 (day) to 700 m (night) and their top altitude ranges between 4.5 and 6.5 km (that is, near the freezing level). Note, however, that the larger thickness and occurrence of TMLCs at night than during day might be due to the larger signal-to-noise ratio of the lidar at night than during day[12]. The mean TMLC spatial coverage is about 10% over

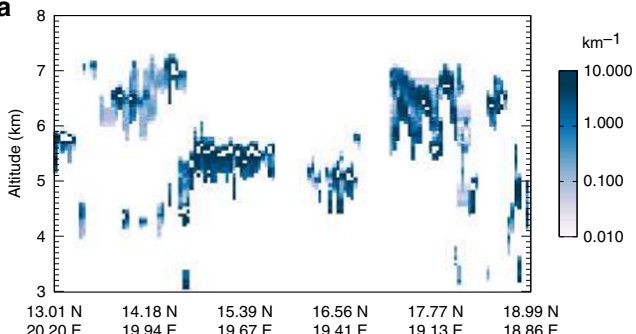

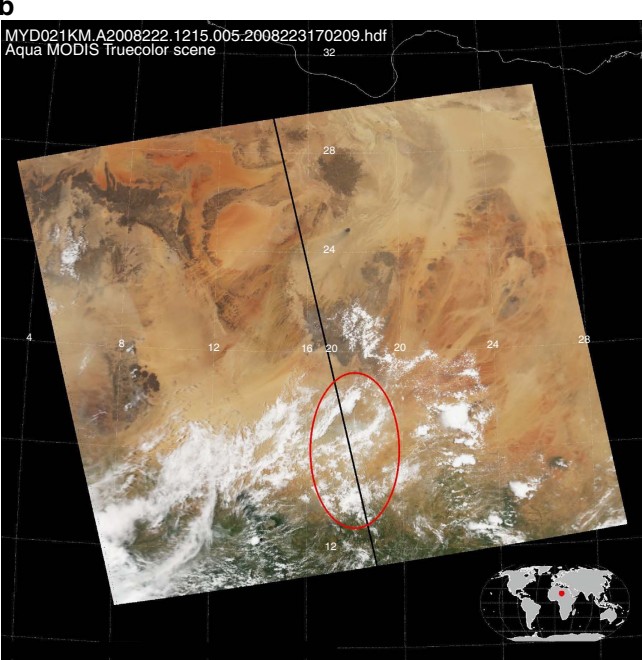

**Figure 1 | Observation of TMLCs. (a)** Extinction of TMLCs as identified in CALIOP retrievals over North Africa in August 2008 and (**b**) its corresponding true colour picture retrieved by the MODIS instrument. Note that both the CALIOP and MODIS instruments on board the CALIPSO and Aqua satellites, respectively, are part of the A-Train constellation meaning that these satellites have the same overpass, separated by 1 min 15 s. (**b**) The black line and red circle represent the orbit track of the satellites and the identified TMLCs, respectively.

tropical land and 5% over tropical ocean indicating that TMLCs are more common over land than over ocean. The annual mean optical depth (OD) of TMLCs in the tropics varies between 0.8 and 1.0 (cloudy sky only), which corresponds to an all-sky (clear sky and cloudy sky) annual mean OD of TMLCs in the tropics of 0.05–0.06 (Fig. 3 and Table 1). The distribution of TMLCs is, however, highly heterogeneous which makes their OD regionally variable (Fig. 3). The all-sky annual mean TMLC OD is 0.10 over land and 0.04 over ocean, and their maximum reaches 0.15 over the Southern African region and 0.09 over the tropical Atlantic Ocean. The TMLC contribution to the tropical annual average cloud OD follows a Gaussian distribution where the TMLC contribution reaches 70–80% between 4 and 7 km. This makes TMLCs the main population of tropical clouds (in terms of OD) within these altitudes under low cloud OD conditions (that is, with a total cloud OD < 5; see Methods).

**Formation processes of TMLCs.** We conducted high-resolution model simulations[15] (see Methods for details) to examine the

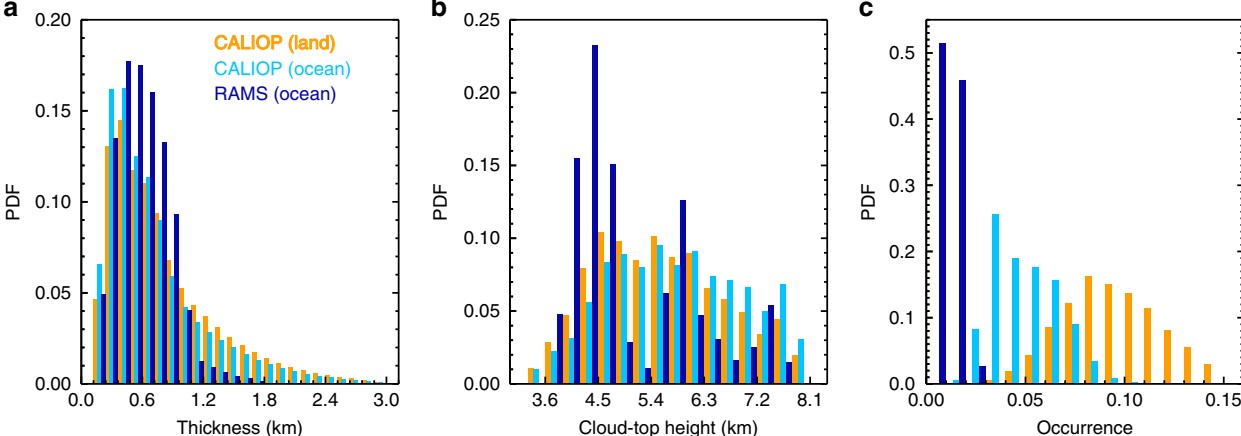

**Figure 2 | Physical properties of TMLCs.** Probability density functions of TMLC thickness (**a**), cloud-top altitude (**b**) and occurrence (**c**) for CALIOP observations in the tropics for the 2008–2012 period over land (orange) and ocean (light blue), and for the high-resolution model (RAMS) results over ocean (dark blue).

**Table 1 | Summary of annual average TMLC physical and optical properties in the tropics as observed by CALIOP for the 2008–2012 time period.**

| | Tropics (land) | | Tropics (ocean) | | Tropics | |
|---|---|---|---|---|---|---|
| | Day | Night | Day | Night | Day | Night |
| Optical Depth* | 1.03 | 0.98 | 0.81 | 0.84 | 0.88 | 0.87 |
| Occurrence in space (%) | 9.8 | 9.2 | 3.7 | 6.3 | 5.1 | 6.9 |
| Optical depth† | 0.10 | 0.09 | 0.03 | 0.05 | 0.05 | 0.06 |
| Thickness (km) | 0.4 | 0.7 | 0.4 | 0.6 | 0.4 | 0.7 |
| Cloud-top altitude (km) | 4.6 | 5.5 | 5.5 | 5.5 | 5.0 | 5.5 |

CALIOP, cloud-aerosol lidar with orthogonal polarization; TMLC, thin mid-level cloud.
*Cloudy sky only.
†All sky (cloudy sky and clear sky).

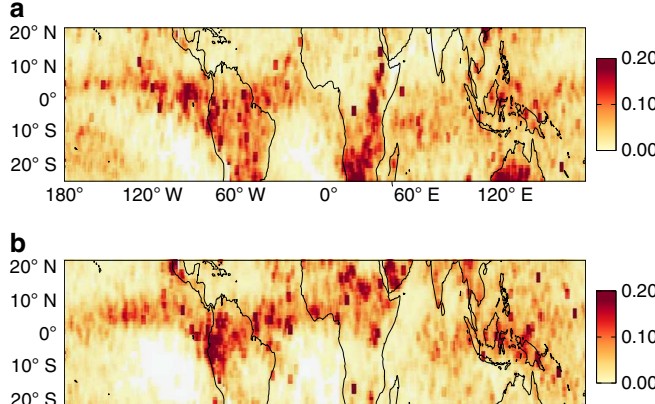

**Figure 3 | Optical depth of TMLCs.** All-sky optical depth of TMLCs in the tropics for January (**a**) and July (**b**) average over 5 years (2008–2012) of CALIOP data.

formation process of TMLCs, and we compared the output from the high-resolution model with the satellite observations. TMLCs are suggested to form in connection with deep convection. The latent heat release associated with the intense freezing and melting near the zero isotherm (~5 km height) during a first convective cycle may generate a shallow temperature inversion, which slows down the vertical motion during subsequent convective cycles, generating convergence and detrainment of moisture near the freezing level[5,7,11]. The detrained moisture may saturate the air and form TMLCs that are transported away horizontally from the core of the deep convective clouds. This potential formation process is consistent with observations (Fig. 3) showing that TMLCs follow the seasonal migration of the Intertropical Convergence Zone (ITCZ). The clouds are mostly found north of the equator during the northern hemisphere summer and south of the equator during the southern hemisphere summer. The modelled TMLCs have similarities with the satellite observations. They are found between 3.5 and 7.5 km with a maximum between 4 and 5 km (Fig. 2). The modelled TMLCs have a mean thickness of 600 m and they are mostly liquid <6.5 km. However, the spatial coverage of the TMLCs is much larger in the CALIOP observations (5%) than in the model simulations (1%), indicating that the detrainment of moisture in the model may be too weak. In models with coarser resolution, including the ERA-Interim Reanalysis data and models from the Coupled Model Intercomparison Project (CMIP5), TMLCs are not reproduced (not shown). This result is perhaps not surprising as these models lack explicit treatment of deep convection and associated detrainment processes.

**Radiative effect of TMLCs.** Since even the high-resolution cloud model cannot reproduce TMLC properties accurately, cloud OD values from satellite data and a simplified radiative transfer model[16] were used to estimate the radiative effect of TMLCs. The reported accuracy of this model to calculate cloud radiative effects is better than 20% compared with a more detailed radiative transfer model[17]. In the following, reported cloud radiative effect values have thus an associated uncertainty of 20%. Although we focus our study on the radiative effects of TMLCs, their influence on heat and moisture transport in the tropics could potentially be at least as important[7]. An annual average TMLC OD of 0.88, an occurrence of 6%, a surface albedo of 0.09, a surface temperature of 300 K and a TMLC temperature of 273 K were used as representative values for TMLCs in the tropics (see Methods). Given these values, and compared with clear-sky conditions, the top of the atmosphere (TOA) net average tropical radiative effect of TMLCs is $-0.6\,W\,m^{-2}$ (shortwave and long-wave components are $-2.4\,W\,m^{-2}$ and $1.8\,W\,m^{-2}$, respectively), inducing a cooling effect (Fig. 4). Note that a variation of the TMLC OD only induces a change in magnitude of the radiative effect, while a variation of the cloud temperature (that is, altitude) changes the sign of the radiative effect from negative to positive when TMLCs are found below a temperature of 263 K (Fig. 4a). However, due to the heterogeneous distribution of TMLCs and the

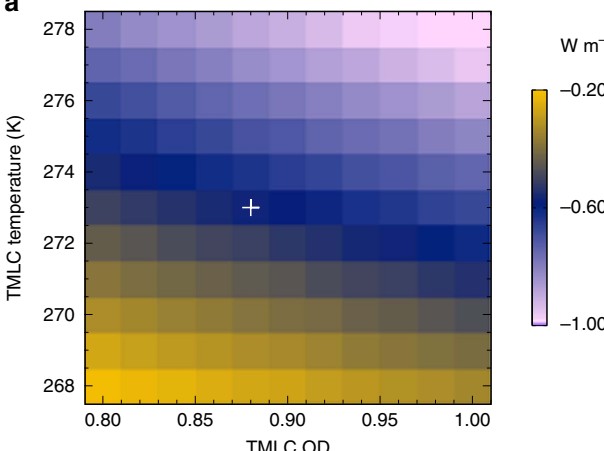

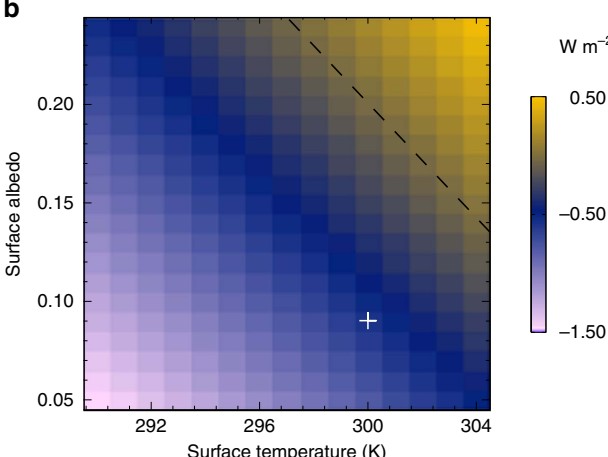

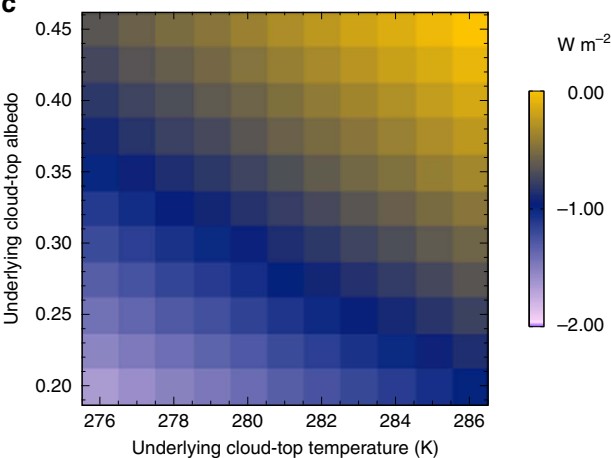

**Figure 4 | Radiative properties of TMLCs.** Dependence of TMLC radiative effect on (**a**) TMLC properties (default, surface temperature = 300 K and surface albedo = 0.09), (**b**) surface properties (default TMLC OD = 0.88 and temperature = 273 K) and (**c**) low-level cloud properties (default, TMLC OD = 0.88 and temperature = 273 K). The white cross indicates the net radiative effect of TMLCs for characteristic tropical values (that is, with a mean TMLC OD of 0.88, TMLC temperature of 273 K, spatial coverage of 6%, surface albedo of 0.09 and surface temperature of 300 K). The dashed line indicates a zero radiative effect.

varying surface temperature and albedo, the radiative effect may vary from one region to another (Fig. 4). For constant TMLC properties and if the surface temperature and albedo are colder and

lower, respectively, than what is shown by the dashed line in Fig. 4b, the radiative effect of the TMLCs would be more negative and *vice versa*. Over bright surfaces such as deserts (albedo $\sim 0.40$), TMLCs have a warming effect while they have a cooling effect over dark surfaces such as oceans (albedo $\sim 0.06$). The diurnal cycle in the physical properties of TMLCs, with a colder temperature (that is, a higher cloud top) and larger coverage during night than during day, also implies that the long-wave radiative effect of TMLCs is larger by about $0.7 \, \text{W m}^{-2}$ during night than during day. This difference could affect the diurnal cycle of temperature and stability in the lower and middle troposphere.

The CALIOP observations indicate that a cloud layer is often present below TMLCs with an annual occurrence of 38%. Clouds below TMLCs are predominantly shallow cumulus formed on top of the active mixing layer (that is, with a top altitude of about 2–3 km)[5]. By considering a cloud layer below the TMLCs with a top temperature ranging from 276 to 286 K and an albedo ranging from 0.20 to 0.45 (see Methods), TMLCs are found to have a TOA net radiative effect varying between $-1.6$ and $0.0 \, \text{W m}^{-2}$ (Fig. 4c). The combination of relatively cold low-level clouds with low albedo below the TMLCs thus induces a stronger negative radiative effect of the TMLCs compared with clear sky. In contrast, TMLCs can also be radiatively obscured by optically-thin high clouds (TMLCs are not observed by CALIOP below opaque clouds). The CALIOP observations indicate that high-level clouds are often present above TMLCs with an annual occurrence of 64%. It makes sense to frequently find TMLCs below optically-thin high clouds as the formation of both cloud types is closely linked to deep convection. The shortwave radiation reaching TMLCs may thus often be reduced by an overlaying cloud layer. If we consider that overlaying clouds have a mean OD of about 1 and a temperature of 235 K (ref. 16), the negative radiative effect of TMLCs would be reduced by $\sim 0.1 \, \text{W m}^{-2}$ according to the cloud radiative forcing model.

Including the impact of finding an additional cloud layer below (38%) and above (64%) TMLCs, the tropical average TOA net radiative effect of TMLCs was calculated to range between $-0.9$ and $-0.3 \, \text{W m}^{-2}$, that is, the clouds induce a cooling. However, a warming effect is obtained over some areas. The largest warming is found for the Saharan desert region during the West African monsoon (that is, when the ITCZ is over this region). In this case, the net radiative effect of TMLCs varies between $-0.2$ and $1.0 \, \text{W m}^{-2}$. The strongest cooling effect is found over the tropical Atlantic Ocean where the net radiative effect of TMLCs ranges between $-1.1$ and $-0.5 \, \text{W m}^{-2}$. Since the net radiative effect of all clouds is about $-20 \, \text{W m}^{-2}$ in the tropics[18], the TMLC contribution is on average about 1.5–4.5% to the overall net cloud radiative forcing in the tropics. This is comparable in magnitude to the global absolute contribution from cirrus (4%)[3].

## Discussion

In a context of a global warming of 2 K (see Methods), a perturbed high-resolution model simulation indicates that the morphological properties of TMLCs, as well as their cloud water content, and thus their OD, remain similar. However, it is difficult to say how TMLCs would change in a changing climate as not even the high-resolution model is able to accurately represent them. The cloud feedback and cloud adjustment to increasing $CO_2$ would likely change the physical and optical properties of clouds at different levels[1,4,19] affecting the properties and radiative effect of TMLCs. Fewer, higher and thicker clouds are simulated to form in a warmer climate[1]. Moreover, their albedo may decrease in the tropics due to the cloud adjustment to increasing $CO_2$ (ref. 19). On the one hand, thicker TMLCs and a decrease of low-level cloud albedo will likely induce a

more negative radiative effect (Fig. 4). On the other hand, fewer clouds and an increase in surface albedo, for example, due to deforestation[20], will lead to a less negative radiative effect (Fig. 4). Changes in the intensity and distribution of deep convection in a warmer climate may also directly affect the TMLC formation process and spatial distribution. All in all, a proper representation of TMLC formation in global climate models is needed to draw more firm conclusions on how the radiative effect of TMLCs would be affected in a warming climate. Our results suggest that further studies on TMLCs are needed to better understand their life cycle and physical properties, to represent them accurately in models, and to fully explore their radiative effects and overall impact on climate.

## Methods

**Satellite observations.** CALIOP version 3 level 2 532 nm cloud extinction data[12] were used in the tropical troposphere. The data were filtered to only take into account clouds between 3 and 8 km. Clouds with their cloud-base or cloud-top height outside this altitude range were rejected. For multi-level cloud layers along the atmospheric column, a minimum threshold of 300 m was set to distinguish TMLCs from low- and high-level clouds. As a consequence, clouds between 3 and 8 km are not considered as TMLCs if they are located < 300 m from another cloud layer. Vertical profiles with a completely attenuated signal in the free troposphere were rejected to account for possible TMLCs. In addition, this rejection procedure prevents the retrieval of cloud extinction values from the cloud-top of cumulus congestus that could completely attenuate the CALIOP signal below their cloud-top and thus could be misclassified as TMLCs. The rejected cloud scenes are not taken into account in the calculation of the occurrence meaning that the occurrence of TMLCs represents the per cent of the tropics that is covered by this cloud type and not covered by the rejected cloud scenes. Note that the CALIOP lidar retrieves aerosol and cloud OD values up to 5 (ref. 12). As a consequence, we are able to calculate the OD contribution of TMLCs for an atmosphere without thick clouds (that is, with an OD < 5). Uncertainties on the OD are within a factor of 2 (ref. 12).

**Model simulations.** The cloud resolving model simulation was performed using the Regional Atmospheric Modeling System (RAMS)[15]. The simulation was conducted at 1 km horizontal grid spacing with 2,000 by 400 points and 65 stretched vertical levels (including 11 in the boundary layer). The vertical resolution at the height of TMLCs averages 325 m. A fixed sea surface temperature of 301 K was used throughout the domain. RAMS includes eight hydrometeor types. All microphysical calculations were performed using the RAMS double-moment parameterizations. The model was initialized from the DYNAMO mean sounding[21] with small horizontally-varying, random perturbations made to the moist potential temperature field to coax convection. Forward integration was done for 60 days and the model achieves a quasi-equilibrium state after ~ 30 days. The final 10 days of the simulation were analysed[22]. A perturbation simulation was also conducted in which the ocean surface temperature was abruptly decreased by 2 K. It allows for a simple and computationally efficient test of simulation sensitivity to model parameters (in this case surface conditions) without the need for running the model for another 60 days. By considering the opposite effect of the decrease of the sea surface temperature by 2 K, we can qualitatively evaluate the impact of a global warming of 2 K on the TMLCs. TMLCs were identified objectively from the simulation output. A total condensate threshold of $0.01 \, g \, kg^{-1}$ (mixing ratio) was applied to identify clouds. Cloudy layers between 3 and 8 km overlain and underlain by at least one model layer of clear air were defined as TMLCs. The phase of the clouds was determined by the depth-weighted cloudy-layer per cent liquid or ice.

**Radiative forcing calculations.** The simplified model for cloud radiative forcing[16] has an accuracy better than 20% compared with a more detailed radiative transfer model[17]. Note that this model calculates a difference of radiative fluxes between two scenes: with and without the cloud layer examined (here, TMLCs). Given the fact that the model uses albedo and temperature as two proxy parameters to calculate the reflected shortwave and outgoing long-wave radiation, respectively, the surface and low-level clouds below TMLCs are treated similarly in the model. The radiative effect of TMLCs is calculated as follows: first, we compute the radiative effect of TMLCs for two scenes separately, over the surface and over underlying clouds. Second, the two distinct radiative effects are multiplied with the probability to find TMLCs in clear sky (62%) and with an underlying cloud layer (38%). Finally, the sum of these two radiative effects is multiplied with the overall probability to find TMLCs in the tropics (6%, cf., Table 1). We consider that TMLCs have an annual tropical mean OD of 0.88 (Table 1) and temperature of 273 K (that is, near the freezing level). For the assessment of the sensitivity of the TMLC cloud radiative effect to variations in TMLC OD and temperature, we use a range from 0.8 to 1.0 for the TMLC OD and a range in TMLC temperature between 268 and 278 K. In clear-sky conditions, an annual mean tropical surface temperature of 300 K (ref. 23) and albedo of 0.09 (obtained by averaging a fixed ocean surface albedo of 0.06 (ref. 24) and the ESA GlobAlbedo product over land in the tropics[25]) were used. The underlying cloud layer was assumed to have a top temperature ranging from 276 to 286 K (ref. 26), and an albedo ranging from 0.20 to 0.45 (refs 27,28).

**Data availability.** All codes that have contributed to the results reported in this study are available on request. CALIOP data have been retrieved through the ICARE Data Services and Center (http://www.icare.univ-lille1.fr). The RAMS source code is available at http://reef.atmos.colostate.edu/ ~ sue/vdhpage/rams.php. The cloud radiative forcing model is available online at http://www.iac.ethz.ch/url/crf/.

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

## Acknowledgements

This work is supported by the Swedish National Space Board (Rymdstyrelsen). M.R.I. is supported by the National Science Foundation under Award No. 1433164. We thank T. Corti and T. Peter for their help with the cloud radiative forcing model. We acknowledge the ICARE Data Services and Center for providing access to the CALIOP data used in this study and tools to process them.

## Author contributions

R.K. had the original idea. Q.B., A.M.L.E. and R.K. designed the study. Q.B. and M.R.I. performed the data analysis and model simulations. Q.B. and A.M.L.E. wrote the manuscript. All authors contributed to the interpretation of the results and commented on the manuscript.

## Additional information

**Competing financial interests:** The authors declare no competing financial interests.

