## [Peer Review File · Nature Communications]

Reviewers' comments:

Reviewer #1 (Remarks to the Author):

This paper presents an observational and numerical study of thin mid-level clouds (TMLCs) to show that such clouds are common in the tropics and formed due to detrainment processes from deep convective clouds that are not well-resolved in coarser grid models. The datasets used (CALIOP), and the methodology of the numerical simulations (RAMS) are well-known and appropriate for such a study. The novelty of the study lies in the application of such datasets to the study of mid-level clouds, which this reviewer would agree are largely "forgotten". The relative absence of studies into mid-level clouds is in some ways surprising. Such a census of these clouds may indeed provoke researchers to take these clouds more seriously.

My concern with the article is that it arguably oversells itself with regards to the radiative impact of such clouds in a way that seems forced. The magnitude of the net radiative forcing is calculated to be of order -1 W/m^2 . The net radiative forcing by all clouds is estimated to be -20 W/m^2 , and regionally SW and LW radiative forcing can have magnitudes $>100 \text{ W/m}^2$ (See Ref. 16). Given mid-level clouds lie at the interface between net cooling below and net warming higher up, perhaps it is not surprising that their net effect is small.

Further, the final paragraph attempts to make a link of changing radiative forcing by TMLCs in response to climate change and deforestation. A weak case is made in Figure 4, but only by way of direct radiative transfer calculations, not calculations within a dynamic model that take into account the response of TMLCs themselves to any possible changes.

I believe a climatology of TMLCs is important, as is a study of the dynamic processes that lead to their formation. What is not clear is that such clouds are of marked radiative significance, particularly when coarse grid models already struggle to accurately reproduce clouds with much heftier radiative signatures. What I might anticipate being of greater significance is an examination of the role such clouds play in the redistribution of energy and moisture in the tropical troposphere, and how their formation modifies precipitation and thermodynamic profiles in the tropics.

Reviewer #2 (Remarks to the Author):

Summary:

In this paper the authors identify, characterize, and estimate the top-of-atmosphere (TOA) radiative impact of an under-studied aspect of the climate system: thin midlevel clouds (TMLCs). They use a cloud resolving model to investigate the formation mechanisms of this cloud type and use a simplified radiative transfer model to estimate the cloud radiative effect (CRE) of this TMLCs, along with its dependence on underlying surface and cloud properties. They estimate an overall small radiative cooling effect of TMLCs of -0.8 to -0.1 Wm^{-2} , averaged over the tropics, which is comparable to (but of opposite sign to) the net radiative impact of cirrus clouds. Some implications of changes in TMLC properties (and changes in their environment) under global warming are discussed, and the authors suggest that more research should be focused on understanding these clouds, given their potential for inducing feedbacks and their under-representation in global models.

Given the lack of attention that has been given to TMLCs, both in the mean climate and in response to climate change, I find the paper to be very novel, of interest to others in the field, and potentially influential in motivating efforts to improve the representation of this cloud type in models. The paper is well written with clear figures. I have, however, many questions regarding the radiative calculations that should be addressed before I can recommend publication. These major concerns, as well as several minor concerns are detailed below.

Major Comments – Radiative calculations:

- There is here is no consideration of how often this cloud type is obscured (radiatively) by overlying clouds. Even though Calipso can “see” these clouds if found below an upper level cloud (assuming the upper level cloud is thin enough), their radiative impact will be highly dependent on the amount and properties of overlying clouds. Also, it seems necessary to mention this in the section discussing climate change (lines 127-136), as changes in the amount and properties of high clouds will determine the relevance of these clouds for cloud feedback.
- A casual reader of this paper will assume when you state that you used a model to calculate CRE that it is a radiative transfer model like RRTM or Fu-Liou. But what you are using is a highly simplified model. This should be made more clear when the model is first mentioned. How confident are you in the CRE values derived using this model? There is a noteworthy lack of uncertainty bounds on the radiative calculations, which seems a bit egregious given the simplicity of the simple model.
- It was unclear to me whether all of these calculations assume completely overcast skies, both for the TMLCs and for the underlying clouds. For example, is the -0.3 Wm^{-2} cooling noted on line 94 the product of the fractional coverage of TMLCs times their overcast CRE? How is the fractional coverage of TMLCs and of underlying clouds treated in Figure 4?
- It would be helpful to see how the TMLC CRE is split between LW vs. SW components.
- It would be helpful to add another matrix to Figure 3 that assesses the sensitivity of TMLC cloud radiative effect to variations in TMLC optical depth and cloud top temperature, assuming some tropical mean (or TMLC-weighted mean) value of underlying cloud and surface properties. Are these clouds in a regime where a 1-sigma change in their OD changes the sign of their CRE? Are these clouds in a regime where a 1-sigma change in their CTT changes the sign of their CRE?
- There are notable differences in TMLC properties between daytime and nighttime. First, do you trust that these are real differences, or could it be an observational artifact? Second, assuming these are real differences, you could calculate the radiative impact of these differences. For example, TMLCs are higher, thicker, and more frequent at night. This suggests that their LW trapping effect (the only effect operating at night) is larger than during the day; could TMLCs potentially impact the diurnal cycle of temperature?
- Lines 131-132: Given that decreasing underlying albedo tends to make TMLC CRE more negative, I don't understand why decreases in underlying cloud fraction would increase the net CRE of TMLCs. Is this a typo?
- Figure 4: To avoid confusion, I suggest inserting the word “underlying” before “cloud” in the x- and y-axis labels of the right panel. As mentioned above, I strongly recommend showing the dependence of TMLC CRE on TMLC properties (CTT and OD) and on overlying cloud properties.

Minor Comments:

- Line 44: “more frequently” than what?

- Lines 53-56: It was not clear to me that you were discussing averages (as listed in Table 1), not the distributions shown in Figure 2.
- Line 64: I don't really understand what is meant by "the OD of a single TMLC is about 1", or why I should care about this one number.
- Line 86, lines 165-168, and Supplementary Figure 1: While I do not doubt your conclusion that models cannot reproduce this cloud mode, it should be mentioned that you are not comparing apples with apples. Ideally, one would use output from the Calipso simulator that is run in many CMIP5 models, and perform the same analysis as is done with the observed Calipso data. This would rule out the ambiguity that comes with comparing products of satellite retrievals with model diagnostics.
- Line 116: Probably more accurate to discuss this in terms of the African monsoon.
- Line 118: "of the all" should be "of all the", I believe
- Line 119: Should be "0.5% - 4%" • Lines 127-128: To me, this sentence seems like a bit of a stretch. The effect of deforestation on the CRE of TMLCs must be tiny given how small their CRE is, how small the change in underlying albedo must be (even if it is large at the surface, tropical rainforests are cloudy regions), and the small area in which it is occurring. In contrast, one thing that may be worth mentioning is the potentially important role of rapid mid-level cloud adjustments to carbon dioxide (Colman and McAvaney, 2011; Wyant et al., 2012; Zelinka et al., 2013). These adjustments occur in response to the change in radiative cooling profile due to CO₂, and can significantly modulate the magnitude of radiative forcing by CO₂. Zelinka et al (2013) quantified that rapid reductions in mid-level clouds in CMIP5 models enhanced the downwelling TOA net radiation by 0.5 W m⁻², which is equal to the amount from high and low clouds combined. This occurs despite the fact that models likely underestimating the amount of TMLCs.
- Line 129: Suggest clarifying that "increase" means "become less negative"
- Line 141: rephrase to "to take into account clouds"
- Lines 142-144: I don't understand what is meant by this statement.
- Line 145: Suggest replacing "parameter" with "criterion"
- Line 161: I don't understand why a +2 K simulation is not performed if that is the signal that is discussed.
- Figure 1: it would be helpful to include in the caption the OD, thickness and altitude of the clouds in this example. Is this a typical TMLC?

References

- Colman R, McAvaney B (2011) On tropospheric adjustment to forcing and climate feedbacks. *Climate Dyn.* 36:1649-1658.
- Wyant MC, Bretherton CS, Blossey PN, Khairoutdinov M (2012) Fast cloud adjustment to increasing CO₂ in a superparameterized climate model. *J. Adv. Model. Earth Syst.* 4.
- Zelinka MD, Klein SA, Taylor KE, Andrews T, Webb MJ, Gregory JM, Forster PM (2013) Contributions of Different Cloud Types to Feedbacks and Rapid Adjustments in CMIP5. *Journal of Climate* 26:5007-5027.

Reviewer #3 (Remarks to the Author):

This paper uses CALIOP measurements to produce a tropical climatology of thin mid level clouds and show that these clouds are in fact quite prevalent in the tropics, particularly over land. Radiative transfer calculations from typical thin midlevel cloud properties are used to calculate the TOA radiative effect and explore potential changes in a warming climate. The authors also use a high-resolution model to support the hypothesis that the clouds are formed from increased detrainment due to stable layers at the melting level, and show that coarser-resolution global models are not able to capture this mechanism.

The comprehensive view of the radiative impact of these clouds and their formation mechanism is quite unique and worth publishing. As the authors state, there has not been a comprehensive statistical view of these clouds in the past. The paper is well-written and gives clear, quantitative results to put these clouds into context. Given that the impact is quite large, I think this needs to be considered in climate simulations and should start an interesting discussion. I was also quite interested in the difference between land and ocean frequency from the observations, as this may shed additional light into the formation processes of the clouds given that land and ocean produce convection of quite different properties.

Overall the method is quite solid. The authors use a robust observational data set. I recommend the paper for publication after the authors respond to the following comments:

1. In the abstract the Author's state that "TMLCs in the tropics have a cooling effect on climate of a similar magnitude (but opposite sign) as cirrus". I didn't find this statement specifically backed up in the main text. The authors give numerical values for the radiative effect of TMLCs, but not for that of cirrus. It would be helpful to specifically support this statement with a reference.
2. On line 55, the authors state that the overall OD of a single TMLC is about 1. This seems quite high compared to all other values given for TMLC throughout the rest of the paper. Is this correct? If so it needs more explanation.
3. How sensitive are the results to the optical depth of the TMLC? The sensitivity tests with albedo and surface temperature are quite useful. It appears from the method section that the authors use a single OD value for the radiative transfer calculations.

Specific comments:

Line 29: I am not sure primordial is the right word

Line 48: The difference in thickness between day and night may be an indication of the higher signal to noise ratio in the lidar data at night than during the day. It might be worth mentioning that this difference may not be a difference in cloud properties but in instrument sensitivities.

Figure 3: I assume the average OD is calculated only when clouds are present, is this true?

Figure S1: Are the 2 and 3 km numbers correct in this figure caption? I don't understand why this was done. I would have thought that only opaque clouds in the free troposphere were removed so that CALIOP observations attenuated above 3 km were not confused with thin clouds.

**Reviewer #1 (Remarks to the Author):**

**This paper presents an observational and numerical study of thin mid-level clouds (TMLCs)**
**to show that such clouds are common in the tropics and formed due to detrainment processes**
**from deep convective clouds that are not well-resolved in coarser grid models. The datasets**
**used (CALIOP), and the methodology of the numerical simulations (RAMS) are well-known**
**and appropriate for such a study. The novelty of the study lies in the application of such**
**datasets to the study of mid-level clouds, which this reviewer would agree are largely**
**"forgotten". The relative absence of studies into mid-level clouds is in some ways surprising.**
**Such a census of these clouds may indeed provoke researchers to take these clouds more**
**seriously.**

We would like to thank the referee for these positive words as well as the review. We very much
appreciated the suggestions and comments that helped us significantly improve the manuscript.

**My concern with the article is that it arguably oversells itself with regards to the radiative**
**impact of such clouds in a way that seems forced. The magnitude of the net radiative forcing**
**is calculated to be of order -1 W/m². The net radiative forcing by all clouds is estimated to be**
**-20 W/m², and regionally SW and LW radiative forcing can have magnitudes >100 W/m² (See**
**Ref. 16). Given mid-level clouds lie at the interface between net cooling below and net**
**warming higher up, perhaps it is not surprising that their net effect is small.**

We agree with the reviewer that it is not surprising that the radiative effect of TMLCs is found to be
relatively small, ranging from -0.9 to -0.3 W.m⁻², since the clouds are found at the border region
between low-level clouds with an overall net cooling effect and high-level clouds (cirrus) with a
warming effect in general. We argue, however, that the radiative forcing of TMLCs is not
insignificant and worth studying since its magnitude is in a range comparable to other clouds such

as cirrus (but of opposite sign) or aerosol particles. Nevertheless, we agree with the reviewer that
we should not oversell our message of the potential radiative effect of TMLCs and have therefore
slightly modified the abstract to accommodate the comment by the reviewer.

**Further, the final paragraph attempts to make a link of changing radiative forcing by TMLCs**
**in response to climate change and deforestation. A weak case is made in Figure 4, but only by**
**way of direct radiative transfer calculations, not calculations within a dynamic model that**
**take into account the response of TMLCs themselves to any possible changes.**

We agree with the reviewer that the study would benefit from calculations of the radiative effect of
TMLCs using a fully dynamic model, however, as indicated in the manuscript, climate models are
currently not able to reproduce TMLCs. Indeed, even the high-resolution model RAMS has
problems representing the TMLC formation processes. As a consequence, we used direct radiative
transfer calculations from a simplified cloud radiative forcing model. The reason for using this
simplified model is now more clearly stated in the manuscript. The authors of the simplified
radiative transfer model indicate that the accuracy of their model is better than 20% compared to the
more detailed Fu and Liou radiative transfer model. We think that this is sufficient as a first step to
estimate the potential radiative effect of TMLCs and to draw attention to TMLCs, especially within
the climate modeling community.

**I believe a climatology of TMLCs is important, as is a study of the dynamic processes that**
**lead to their formation. What is not clear is that such clouds are of marked radiative**
**significance, particularly when coarse grid models already struggle to accurately reproduce**
**clouds with much heftier radiative signatures. What I might anticipate being of greater**
**significance is an examination of the role such clouds play in the redistribution of energy and**
**moisture in the tropical troposphere, and how their formation modifies precipitation and**

**thermodynamic profiles in the tropics.**

Once again, we agree with the reviewer and thank him/her for this remark. Using the same high-
resolution cloud model as we have used (RAMS), Iwasa et al. [2012] showed that TMLCs may very
well be important for heat and moisture transport in the tropics. This is now also pointed out in the
manuscript. However, as even the high-resolution model RAMS has problems simulating TMLCs,
it is difficult, if not impossible, to carefully investigate the role of TMLCs for energy and moisture
transport, precipitation, etc. This is also the reason why we think it is important to point out the
existence of TMLCs to the climate community and hope that this study will be influential in
motivating future work towards including the processes relevant for TMLC formation in models.
We would also like to stress once more that even if the radiative effect of TMLCs appears to be
relatively small, it could still be about the same order of magnitude as the radiative effect of cirrus
or aerosol particles.

**Reviewer #2 (Remarks to the Author):**

**Summary:**

**In this paper the authors identify, characterize, and estimate the top-of-atmosphere (TOA)**

**radiative impact of an under-studied aspect of the climate system: thin mid-level clouds**

**(TMLCs). They use a cloud resolving model to investigate the formation mechanisms of this**

**cloud type and use a simplified radiative transfer model to estimate the cloud radiative effect**

**(CRE) of this TMLCs, along with its dependence on underlying surface and cloud properties.**

**They estimate an overall small radiative cooling effect of TMLCs of -0.8 to -0.1 Wm⁻²,**

**averaged over the tropics, which is comparable to (but of opposite sign to) the net radiative**

**impact of cirrus clouds. Some implications of changes in TMLC properties (and changes in**

**their environment) under global warming are discussed, and the authors suggest that more**

**research should be focused on understanding these clouds, given their potential for inducing**

**feedbacks and their under representation in global models.**

**Given the lack of attention that has been given to TMLCs, both in the mean climate and in**

**response to climate change, I find the paper to be very novel, of interest to others in the field,**

**and potentially influential in motivating efforts to improve there presentation of this cloud**

**type in models. The paper is well written with clear figures. I have, however, many questions**

**regarding the radiative calculations that should be addressed before I can recommend**

**publication. These major concerns, as well as several minor concerns are detailed below.**

**We would like to thank the referee for these positive words as well as the careful review. We very**

**much appreciated the suggestions and comments that helped us significantly improve the**

**manuscript.**

**Major Comments – Radiative calculations:**

• **There is here no consideration of how often this cloud type is obscured (radiatively) by**
**overlying clouds. Even though Calipso can “see” these clouds if found below an upper level**
**cloud (assuming the upper level cloud is thin enough), their radiative impact will be highly**
**dependent on the amount and properties of overlying clouds. Also, it seems necessary to**
**mention this in the section discussing climate change (lines 127-136), as changes in the amount**
**and properties of high clouds will determine the relevance of these clouds for cloud feedback.**

We thank the reviewer for pointing this out, it is a very good suggestion. We have now included
calculations of the radiative effect of TMLCs when clouds are found above and below them.

According to Corti and Peter [2009], high clouds typically have a mean OD of 1 and a temperature
of 235 K. This implies a reduction of the incoming shortwave solar radiation by about 35 W.m^{-2} . If
the cirrus clouds are found above TMLCs with an albedo of 0.2, this could reduce their negative
radiative effect by 0.1 W.m^{-2} .

• **A casual reader of this paper will assume when you state that you used a model to calculate**
**CRE that it is a radiative transfer model like RRTM or Fu-Liou. But what you are using is a**
**highly simplified model. This should be made more clear when the model is first mentioned.**
**How confident are you in the CRE values derived using this model? There is a noteworthy**
**lack of uncertainty bounds on the radiative calculations, which seems a bit egregious given the**
**simplicity of the simple model.**

According to the remark by the reviewer, we now explain in the main text that we use a simplified
radiative transfer model. We also mention that the model has been tested and evaluated, and that the
authors report an accuracy in CRE better than 20% compared to the Fu and Liou radiative transfer
model. In order to keep the manuscript clear, we mention this uncertainty on CRE but do not
include the associated uncertainty on all values since it is always 20%.

• **It was unclear to me whether all of these calculations assume completely overcast skies, both**
**for the TMLCs and for the underlying clouds. For example, is the -0.3 Wm^{-2} cooling noted on**
**line 94 the product of the fractional coverage of TMLCs times their overcast CRE? How is the**
**fractional coverage of TMLCs and of underlying clouds treated in Figure 4?**

It is true that the difference between cloudy-sky and all-sky OD of TMLCs was not well described
and it has been improved in the new version of the manuscript. In the previous version of the
manuscript, we were using the all-sky OD of TMLCs for the calculation of radiative effects while
we should have used the cloudy-sky OD of TMLCs multiplied by a coefficient: the occurrence in
space or surface coverage of TMLCs. We have changed our method section of the manuscript and
the way we calculated the radiative effect of TMLCs. As a consequence, the mean radiative effect of
TMLCs is now more negative than before.

• **It would be helpful to see how the TMLC CRE is split between LW vs. SW components.**

We now state the shortwave and longwave components of the TMLC CRE for the characteristic
tropical value conditions: -2.4 and 1.8 W.m^{-2} , respectively.

• **It would be helpful to add another matrix to Figure 4 that assesses the sensitivity of TMLC**
**cloud radiative effect to variations in TMLC optical depth and cloud top temperature,**
**assuming some tropical mean (or TMLC-weighted mean) value of underlying cloud and**
**surface properties. Are these clouds in a regime where a 1-sigma change in their OD changes**
**the sign of their CRE? Are these clouds in a regime where a 1-sigma change in their CTT**
**changes the sign of their CRE?**

We thank the reviewer for pointing this out, it is a very good suggestion. We added a third plot in
Figure 4 representing the sensitivity of the TMLC radiative effect to variations in TMLC OD and

temperature. This plot shows that even a TMLC OD ranging from 0.8 to 1.0 (which are the mean
TMLC OD over ocean and land, respectively) and a change of 5 K of the TMLC temperature do not
change the sign of the CRE and it remains negative. We did not use a 1-sigma OD range in Figure
4a because the variability of TMLC OD is naturally large (about 0.8 which is similar to their mean
OD). The TMLC OD is very likely proportional with the “age” of the cloud. As a consequence, new
formed TMLCs likely have a larger OD than “old” (almost dissipated) TMLCs. However, the
“new” and “old” TMLCs are balanced and their mean OD is about 0.88. In contrast, the TMLC
temperature is much more centered near the 273 K (between 4 and 5.5 km, cf. Figure 2).

• **There are notable differences in TMLC properties between daytime and nighttime. First, do**
**you trust that these are real differences, or could it be an observational artifact? Second,**
**assuming these are real differences, you could calculate the radiative impact of these**
**differences. For example, TMLCs are higher, thicker, and more frequent at night. This**
**suggests that their LW trapping effect (the only effect operating night) is larger than during**
**the day; could TMLCs potentially impact the diurnal cycle of temperature?**

We agree with the reviewer, the difference between daytime and nighttime may be due to a smaller
signal-to-noise during day than at night. This is now mentioned in the text. In addition, we also
discuss the daily difference in physical properties of TMLCs and the potential impact on the LW
trapping effect.

• **Lines 131-132: Given that decreasing underlying albedo tends to make TMLC CRE more**
**negative, I don't understand why decreases in underlying cloud fraction would increase the**
**net CRE of TMLCs. Is this a typo?**

According to CALIOP, TMLCs are found with and without an underlying cloud layer below them
with an occurrence of 38% and 62%, respectively.

Without: TMLC CRE is -0.6 W.m^{-2}

With: TMLC CRE is about -0.8 W.m^{-2} ($-1.6 < \text{TMLC RE} < 0.0 \text{ W.m}^{-2}$)

Total TMLC CRE = $-0.6 * 0.62 - 0.8 * 0.38 = -0.68 \text{ W.m}^{-2}$

If we decrease the amount of low-level clouds and consider that we find them below TMLCs with
an occurrence of 30% (instead of 38%).

Total TMLC CRE = $-0.6 * 0.7 - 0.8 * 0.3 = -0.6 \text{ W.m}^{-2}$

Therefore, the TMLC CRE increases or becomes less negative with a decrease in underlying cloud
coverage fraction.

• **Figure 4: To avoid confusion, I suggest inserting the word “underlying” before “cloud” in**
**the x- and y-axis labels of the right panel. As mentioned above, I strongly recommend showing**
**the dependence of TMLC CRE on TMLC properties (CTT and OD) and on overlying cloud**
**properties.**

We added the word “underlying” before “cloud” in labels of the x- and y-axis. We also show the
dependence of TMLC RE on TMLC properties.

**Minor Comments:**

• **Line 44: “more frequently” than what?**

We have removed the word “more”.

• **Lines 53-56: It was not clear to me that you were discussing averages (as listed in Table 1),**
**not the distributions shown in Figure 2.**

We have clarified that we use distribution and averages from Figure 2 and Table 1, respectively.

• **Line 64: I don’t really understand what is meant by “the OD of a single TMLC is about 1”,**
**or why I should care about this one number.**

Following the previous comment by the reviewer, we now explain the difference between cloudy-

sky and full-sky OD of TMLCs.

• **Line 86, lines 165-168, and Supplementary Figure 1: While I do not doubt your conclusion**
**that models cannot reproduce this cloud mode, it should be mentioned that you are not**
**comparing apples with apples. Ideally, one would use output from the Calipso simulator that**
**is run in many CMIP5 models, and perform the same analysis as is done with the observed**
**Calipso data. This would rule out the ambiguity that comes with comparing products of**
**satellite retrievals with model diagnostics.**

We tried to use the CFMIP Observation Simulator Package (COSP) on the RAMS output that
allows to simulate the signal that CALIOP would see in the RAMS results, however, COSP was
removing most of TMLCs produced in RAMS because it seems that cirrus clouds were not optically
thin enough to let the “CALIOP signal” penetrating them. Therefore, we decided not to use COSP
on RAMS output. However, we now explicitly state in the figure text that comparing model output
and satellite retrievals is not straight-forward.

• **Line 116: Probably more accurate to discuss this in terms of the African monsoon.**

This is now discussed in terms of the West African monsoon.

• **Line 118: “of the all” should be “of all the”, I believe**

Corrected.

• **Line 119: Should be “0.5% - 4%”**

Since we changed the method for the calculation of TMLC radiative effects, this changed the
percentage to 1.5-4.5%.

• **Lines 127-128: To me, this sentence seems like a bit of a stretch. The effect of deforestation**
**on the CRE of TMLCs must be tiny given how small their CRE is, how small the change in**
**underlying albedo must be (even if it is large at the surface, tropical rainforests are cloudy**
**regions), and the small area in which it is occurring. In contrast, one thing that may be worth**
**mentioning is the potentially important role of rapid mid-level cloud adjustments to carbon**

**dioxide (Colman and McAvaney, 2011; Wyant et al., 2012; Zelinka et al., 2013). These**
**adjustments occur in response to the change in radiative cooling profile due to CO₂, and can**
**significantly modulate the magnitude of radiative forcing by CO₂. Zelinka et al (2013)**
**quantified that rapid reductions in mid-level clouds in CMIP5 models enhanced the**
**downwelling TOA net radiation by 0.5 W m⁻², which is equal to the amount from high and**
**low clouds combined. This occurs despite the fact that models likely underestimate the**
**amount of TMLCs.**

We thank the reviewer for pointing this out. We now discuss more in detail the cloud adjustments to
increasing CO₂, and their potential impact on TMLCs.

• **Line 129: Suggest clarifying that “increase” means “become less negative”**

Corrected.

• **Line 141: rephrase to “to take into account clouds”**

Corrected.

• **Lines 142-144: I don’t understand what is meant by this statement**

We changed that sentence for: “For multi-level cloud layers along the atmospheric column, a
minimum threshold of 300 m was set to distinguish TMLCs from low- and high-level clouds. As a
consequence, clouds between 3 and 8 km are not considered as TMLCs if they are located less than
300 m from another cloud layer.”

• **Line 145: Suggest replacing “parameter” with “criterion”**

Corrected for “rejection procedure”.

• **Line 161: I don’t understand why a +2 K simulation is not performed if that is the signal**
**that is discussed.**

In principle, it would be better to increase the temperature rather than to decrease it. Unfortunately,
it takes the atmosphere a long time to equilibrate to a sudden step increase in surface temperature
because it requires a lot of surface evaporation in order to bring the humidity back up. By dropping

the temperature, the atmosphere quickly saturates with existing water which it rains out in order to
return to a relevant, domain-mean relative humidity. So, a drop in temperature allows the
atmosphere to adjust much more quickly to save computational time.

• **Figure 1: it would be helpful to include in the caption the OD, thickness and altitude of the**
**clouds in this example. Is this a typical TMLC?**

The corresponding CALIOP cloud extinction along the chosen orbit track is shown in Figure 1 and
the thickness and altitude of TMLCs in this example is now mentioned in the text.

**References**

• **Colman R, McAvaney B (2011) On tropospheric adjustment to forcing and climate**
**feedbacks. Climate Dyn. 36:1649-1658.**

• **Wyant MC, Bretherton CS, Blossey PN, Khairoutdinov M (2012) Fast cloud adjustment to**
**increasing CO2 in a superparameterized climate model. J. Adv. Model. Earth Syst. 4.**

• **Zelinka MD, Klein SA, Taylor KE, Andrews T, Webb MJ, Gregory JM, Forster PM (2013)**
**Contributions of Different Cloud Types to Feedbacks and Rapid Adjustments in CMIP5.**
**Journal of Climate 26:5007-5027.**

**Reviewer #3 (Remarks to the Author):**

**This paper uses CALIOP measurements to produce a tropical climatology of thin mid level**
**clouds and show that these clouds are in fact quite prevalent in the tropics, particularly over**
**land. Radiative transfer calculations from typical thin midlevel cloud properties are used to**
**calculate the TOA radiative effect and explore potential changes in a warming climate. The**
**authors also use a high-resolution model to support the hypothesis that the clouds are formed**
**from increased detrainment due to stable layers at the melting level, and show that coarser-**
**resolution global models are not able to capture this mechanism.**

**The comprehensive view of the radiative impact of these clouds and their formation**
**mechanism is quite unique and worth publishing. As the authors state, there has not been a**
**comprehensive statistical view of these clouds in the past. The paper is well-written and gives**
**clear, quantitative results to put these clouds into context. Given that the impact is quite large,**
**I think this needs to be considered in climate simulations and should start an interesting**
**discussion. I was also quite interested in the difference between land and ocean frequency**
**from the observations, as this may shed additional light into the formation processes of the**
**clouds given that land and ocean produce convection of quite different properties.**

**Overall the method is quite solid. The authors use a robust observational data set. I**
**recommend the paper for publication after the authors respond to the following comments:**

We would like to thank the referee for these positive words as well as the careful review. We very
much appreciated the suggestions and comments that helped us significantly improve the
manuscript.

**1. In the abstract the Author's state that "TMLCs in the tropics have a cooling effect on**
**climate of a similar magnitude (but opposite sign) as cirrus". I didn't find this statement**
**specifically backed up in the main text. The authors give numerical values for the radiative**
**effect of TMLCs, but not for that of cirrus. It would be helpful to specifically support this**
**statement with a reference.**

Chen et al. [2000] indicates that globally, the global TOA radiative effect of cirrus is 1.3 W.m^{-2}
while the TOA radiative effect of all the clouds is -33.4 W.m^{-2} . Therefore, cirrus account for about
4% of the TOA radiative effect of all the clouds. Furthermore, according to Su et al. [2010], the net
TOA radiative effect of clouds is about -20 W.m^{-2} in the tropics (our region of interest) and we find
that the TOA radiative effect of TMLCs ranges between -0.9 and -0.3 W.m^{-2} (see the updated
method section for how we calculate the radiative effect of TMLCs) which corresponds to 1.5-4.5%
of the TOA radiative effect of all the clouds. Therefore, the radiative effect of TMLCs has the same
order of magnitude as that of cirrus (but of opposite sign). We used relative contribution of cirrus
(4%, instead of 1.3 W.m^{-2}) because we did not find the TOA radiative effect of cirrus in the tropics
only, therefore, we preferred to use the relative contribution of cirrus to the radiative effect of all the
clouds rather than its absolute radiative effect.

**2. On line 55, the authors state that the overall OD of a single TMLC is about 1. This seems**
**quite high compared to all other values given for TMLC throughout the rest of the paper. Is**
**this correct? If so it needs more explanation.**

We thank the reviewer for pointing this out; we agree that the wording was confusing. We now
clearly distinguish the OD of TMLCs by cloudy sky only and all sky (cloudy sky + clear sky) in the
main text, Table 1 and Figure 3. As an annual average, the TMLC OD is 0.88 and 0.05 for cloudy
and all sky, respectively, in the tropics. The relation between these two numbers is obtained by
using the occurrence in space of TMLCs (~6%).

**3. How sensitive are the results to the optical depth of the TMLC? The sensitivity tests with**
**albedo and surface temperature are quite useful. It appears from the method section that the**
**authors use a single OD value for the radiative transfer calculations.**

This is a very good suggestion. We added a third plot to Figure 4 representing the sensitivity of the
TMLC radiative effect to variations in TMLC OD and temperature, and we comment on the results
in the text. Note also that a new method is now used and better documented to calculate the
radiative effect of TMLCs. In this new method, we use the annual mean TMLC OD for cloudy sky
in the tropics (0.88). Then, the resulting radiative effect is multiplied by the occurrence in space to
find TMLCs in the tropics (6%).

**Specific comments:**

**Line 29: I am not sure primordial is the right word**

Changed to "essential".

**Line 48: The difference in thickness between day and night may be an indication of the higher**
**signal to noise ratio in the lidar data at night than during the day. It might be worth**
**mentioning that this difference may not be a difference in cloud properties but in instrument**
**sensitivities.**

The reviewer is right. We now indicate that the difference between day and night might be due to
the larger signal to noise ratio at night than during the day. Nevertheless, in case the day/night
difference in TMLC properties is real, we discuss the diurnal variation in physical properties of
TMLCs and the potential impact on the LW trapping effect.

**Figure 3: I assume the average OD is calculated only when clouds are present, is this true?**

The average TMLC OD is calculated for both cloudy sky only and all sky (cloudy sky + clear sky).

This is now better explained in the main text and in the Figures. Therefore, there are two TMLC

ODs. Cloudy sky OD is 0.88 while it is 0.05 in all sky.

**Figure S1: Are the 2 and 3 km numbers correct in this figure caption? I don't understand why**

**this was done. I would have thought that only opaque clouds in the free troposphere were**

**removed so that CALIOP observations attenuated above 3 km were not confused with thin**

**clouds.**

The 2 and 3 km numbers are correct. In order to distinguish TMLCs from low-clouds reaching

altitudes higher than 3 km, we only consider clouds between 3 and 7 km to be TMLCs and only if

they have at least 300 m of clear air below/above them. Therefore, to remove low clouds reaching

altitudes higher than 3 km in the global model output (in order not to confuse these clouds with

TMLCs), we only consider cloud cover profiles with a cloud cover of less than 10% between 2 and

3 km. We have to acknowledge that this threshold is somewhat arbitrary in magnitude and altitude

but this is done to compensate our observation threshold of 300 m of clear air below TMLCs.

Reviewers' Comments

Reviewer #1 (Remarks to the Author):

The revised version makes it more clear how TMLCs could be radiatively important, should be better acknowledged by the atmospheric science community yet there is the challenge that they cannot even be reproduced in coarse and fine grid models. This last point addresses my concern that dynamic feedbacks make calculation of the future role of such clouds in a changing climate difficult to assess.

As a stylistic note, the final paragraph contains the expression " it is difficult to say" twice.

Reviewer #2 (Remarks to the Author):

Summary:

The authors have addressed many of my comments in a satisfactory manner. However, I still have several remaining concerns, which I detail below.

Major Comments:

- I find it difficult to understand the reported values of spatial cloud coverage, since there are so many scenes that are excluded from analysis (due to obscuration by overlying clouds or by low clouds that extend up into mid-levels). Is the occurrence in space listed in Table 1 and in the text at various points meant to represent the percent of the entire tropical area that is covered by this cloud type, or the percent of the tropics that is covered by this cloud type AND not covered by overlying clouds with OD>5 AND not covered by low clouds that extend into mid-levels? These qualifiers need to be presented more clearly to the reader when coverage statistics are stated.
- I have a number of questions about the calculations and sensitivity studies performed in Figure 4:
 - CRE is generally computed as the difference between all- and clear-sky TOA fluxes, which one can express as the difference between overcast- and clear-sky fluxes, scaled by the cloud fraction. Is the dependence of CRE on underlying cloud properties (Fig. 4c) being computed by taking scenes with overcast TMLC (as well as underlying clouds with some fractional coverage?) and subtracting off scenes with only underlying clouds (with some fractional coverage?), rather than subtracting off a clear-sky scene? Or is the only thing that is modified in this calculation the overcast-sky radiation? Why should the readers trust the simplified model for correctly simulating the upwelling flux from multi-layered clouds? Please explain in more detail how an additional cloud layer is accounted for in the radiation model and what is actually being differenced in order to generate each panel of Figure 4.
 - Given the fact that clear-sky OLR is actually a function of the emission temperature of the scene, which is generally much cooler than the surface, the surface temperature is really just a proxy for this emission temperature (few photons make it from the surface to space). Similarly, the underlying albedo is the property that matters for SWCRE; it should not care whether it is a cloud or

the surface that is reflecting sunlight. Thus, are (b) and (c) fundamentally different? If panel (b) were expanded to include albedos up to 0.45 and underlying temperatures down to 276K, would panel (c) be necessary? If there is a difference, I don't understand why.

- The presence of an underlying cloud should make the negative SW CRE less negative since the albedo contrast is decreased. Thus, the stronger negative radiative effect of TMLC when an underlying cloud is present is apparently coming from a less positive LW CRE. But this large LW effect does not make intuitive sense to me, if LWCRE is computed as the OLR contrast between scenes with and without TMLCs (but both containing low clouds): OLR should not be very sensitive to the presence or absence of low clouds (the bulk of LW emission is coming from the mid-troposphere, well above both the surface and low cloud layer). My confusion probably relates to the fact that I don't actually understand what is being differenced.

Minor Comments:

- Line 17: "ground surface" is awkward, and should probably just be "surface", though either way it is just a proxy for the emission temperature of the clear-sky atmosphere.
- Lines 54-55: "a few hundred meters thick up to two kilometers" should be "between a few hundred meters and two kilometers thick".
- Line 107 "slight" is ambiguous, and it is not clear to the reader that slight variations are realistic. Doesn't the fact that the large natural variability in TMLC OD (mentioned in the response to reviewers) suggest that this is an important effect that should be quantified (rather than using a "slight" variation in OD)?
- Line 112: it is probably worth specifying that this statement assumes constant TMLC properties.
- Line 337: The black cross does not represent the radiative effect of TMLCs but rather the location on the figure where a characteristic TMLC would be located.
- Line 151: the cloud adjustment to CO₂ is considered to be independent of global warming; the portion of cloud changes that occurs in response to global warming is the feedback.
- Line 153: "are believed to" should be "are simulated to".
- Line 154: climate models robustly predict that the albedo of low clouds at middle latitudes will increase. You should probably specify low latitudes if that is what you are referring to.
- Line 157: Suggest "lead to a less negative".
- Lines 351-354: This is a step in the right direction, but still quite inadequate in terms of trying to compare apples and apples. A simulator would be necessary, followed by a similar process of filtering that was applied to observations. I also do not understand why profiles with substantial coverage of overlying high clouds were not excluded.

Reviewer #3 (Remarks to the Author):

I am satisfied with the author's response to my concerns, and recommend the article for publication.

Review of "The forgotten clouds of the tropical middle troposphere"

NCOMMS-16-03257-T (revision 1)

Summary:

The authors have addressed many of my comments in a satisfactory manner. However, I still have several remaining concerns, which I detail below.

We would like to thank the referee for these positive words as well as the review. We very much appreciate the suggestions and comments that helped us significantly improve the manuscript.

Major Comments:

- I find it difficult to understand the reported values of spatial cloud coverage, since there are so many scenes that are excluded from analysis (due to obscuration by overlying clouds or by low clouds that extend up into mid-levels). Is the occurrence in space listed in Table 1 and in the text at various points meant to represent the percent of the entire tropical area that is covered by this cloud type, or the percent of the tropics that is covered by this cloud type AND not covered by overlying clouds with OD>5 AND not covered by low clouds that extend into mid-levels? These qualifiers need to be presented more clearly to the reader when coverage statistics are stated.**

The occurrence of TMLCs represents the percent of the tropics that is covered by this cloud type AND not covered by overlying clouds with OD>5 AND not covered by low clouds that extend into mid-levels. This is now clarified in the method section of the manuscript.

- I have a number of questions about the calculations and sensitivity studies performed in**

Figure 4:

We acknowledge that we did not explain in detail our method to calculate the CRE of TMLCs and we thank the reviewer for pointing this out. The following comments of the reviewer are now answered in the method section of the manuscript.

- CRE is generally computed as the difference between all- and clear-sky TOA fluxes, which one can express as the difference between overcast- and clear-sky fluxes, scaled by the cloud fraction. Is the dependence of CRE on underlying cloud properties (Fig. 4c) being computed by taking scenes with overcast TMLC (as well as underlying clouds with some fractional coverage?) and subtracting off scenes with only underlying clouds (with some fractional coverage?), rather than subtracting off a clear-sky scene? Or is the only thing that is modified in this calculation the overcast-sky radiation? Why should the readers trust the simplified model for correctly simulating the upwelling flux from multi-layered clouds? Please explain in more detail how an additional cloud layer is accounted for in the radiation model and what is actually being differenced in order to generate each panel of Figure 4.

The radiation model applied is quite simplified in that the underlying cloud layer is treated as another type of surface over which TMLCs are found. The CRE is calculated as follows: first, we compute the CRE of TMLCs with the surface and underlying clouds separately, and we multiply each value with the occurrence to find TMLCs (6% in the tropics). Second, since CALIOP retrieves TMLCs in clear-sky (62%) and with an underlying cloud layer (38%), we apply these coefficients on the previously computed CREs and add them to obtain the final CRE of TMLCs.

- Given the fact that clear-sky OLR is actually a function of the emission temperature of the scene, which is generally much cooler than the surface, the surface temperature is really just a proxy for this emission temperature (few photons make it from the surface to space). Similarly, the underlying albedo is the property that matters for SWCRE; it should not care whether it is a cloud or the surface that is reflecting sunlight. Thus, are (b) and (c) fundamentally different? If panel (b) were expanded to include albedos up to 0.45 and

underlying temperatures down to 276K, would panel (c) be necessary? If there is a difference, I don't understand why.

The reviewer is right. The albedo and temperature of the surface and underlying clouds are treated similarly. Therefore, if panel (b) was expanded to include higher albedos and lower temperatures, panel (c) would be included in panel (b). However, we split our plot in two to make it easier for the reader to understand that we consider in panel (b) the surface and in panel (c) an underlying cloud layer, even though they are similarly treated in the model. Moreover, an expanded version of panel (b) would be very large (about 4 times the size of this one) while we would consider only half of it.

- The presence of an underlying cloud should make the negative SW CRE less negative since the albedo contrast is decreased. Thus, the stronger negative radiative effect of TMLC when an underlying cloud is present is apparently coming from a less positive LW CRE. But this large LW effect does not make intuitive sense to me, if LWCRE is computed as the OLR contrast between scenes with and without TMLCs (but both containing low clouds): OLR should not be very sensitive to the presence or absence of low clouds (the bulk of LW emission is coming from the mid-troposphere, well above both the surface and low cloud layer). My confusion probably relates to the fact that I don't actually understand what is being differenced.

As mentioned above, the albedo and temperature of the surface or underlying cloud controls the SW and LW CRE, respectively, for constant cloud properties. Therefore, the presence of an underlying cloud makes the CRE less negative (less negative SW CRE) through an increase in albedo but it makes the CRE more negative (less positive LW CRE) through a decrease in temperature. Overall, it seems that the CRE of TMLCs is slightly more negative with an underlying cloud than over the surface.

Minor Comments:

- **Line 17: “ground surface” is awkward, and should probably just be “surface”, though either way it is just a proxy for the emission temperature of the clear-sky atmosphere.**

Corrected for “surface”.

- **Lines 54-55: “a few hundred meters thick up to two kilometers” should be “between a few hundred meters and two kilometers thick”.**

Corrected for “between a few hundred meters and two kilometers thick”.

- **Line 107 “slight” is ambiguous, and it is not clear to the reader that slight variations are realistic. Doesn’t the fact that the large natural variability in TMLC OD (mentioned in the response to reviewers) suggest that this is an important effect that should be quantified (rather than using a “slight” variation in OD)?**

Changed the sentence for “Note that a variation of the TMLC OD only induces a change in magnitude of their radiative effect while a variation of their temperature (i.e. altitude) changes the sign of their radiative effect from negative to positive when TMLCs are found below a temperature of 264 K”.

- **Line 112: it is probably worth specifying that this statement assumes constant TMLC properties.**

Changed the sentence for “For constant TMLC properties, if the surface temperature and albedo are colder and lower, respectively, than what is shown by the dashed line in Figure 4b, the radiative effect of the TMLCs would be more negative, and vice versa.”.

- **Line 337: The black cross does not represent the radiative effect of TMLCs but rather the location on the figure where a characteristic TMLC would be located.**

Changed “represents” for “indicates”.

- **Line 151: the cloud adjustment to CO₂ is considered to be independent of global warming; the portion of cloud changes that occurs in response to global warming is the feedback.**

Corrected for “The cloud feedback and cloud adjustment to increasing CO₂”.

- **Line 153: “are believed to” should be “are simulated to”.**

Corrected for “are simulated to”.

- **Line 154: climate models robustly predict that the albedo of low clouds at middle latitudes will increase. You should probably specify low latitudes if that is what you are referring to.**

Changed the sentence for “Although the albedo of low-level clouds is predicted to increase due to a changing concentration of cloud condensation nuclei, their albedo may decrease in the tropics due to the cloud adjustment to increasing CO₂.”.

- **Line 157: Suggest “lead to a less negative”.**

Corrected for “lead to a less negative”. Accordingly, we also corrected “larger negative” for “more negative” in the sentence before.

- **Lines 351-354: This is a step in the right direction, but still quite inadequate in terms of trying to compare apples and apples. A simulator would be necessary, followed by a similar process of filtering that was applied to observations. I also do not understand why profiles with substantial coverage of overlying high clouds were not excluded.**

We acknowledge that the comparison between modeled and observed TMLCs is simplified. However, regardless the method of comparison, the models should still show some kind of indication of TMLCs and this is not the case. Note that we now exclude profiles with substantial coverage (>10%) of overlying high clouds.

Reviewers' Comments

Reviewer #2 (Remarks to the Author):

The authors have addressed most of my comments, but a few issues still remain, which I list below. After addressing these comments, I believe the paper will be suitable for publication.

Lines 15-16: "have a cooling effect on climate that could be as large as cirrus (but of opposite sign)." Is awkward. Suggest "have a cooling effect on climate that could be as large in magnitude as the warming effect of cirrus"

Lines 155-158: "Although the albedo of low-level clouds is predicted to increase due to a changing concentration of cloud condensation nuclei, their albedo may decrease in the tropics due to the cloud adjustment to increasing CO₂." This statement needs to be supported with references. Also, the first part of the statement does not reflect the current thinking in this area. Low-level cloud albedo is expected to rise at mid- to high-latitudes owing primarily to warming-induced ice-to-liquid transitions, and secondarily to increases in the adiabatic water content of clouds as the warmer atmosphere becomes more moist. These are not dependent on changing CCN, which may or may not increase with increasing CO₂.

Lines 210-211: I do not understand this statement: "first, we compute the radiative effect of TMLCs with the surface and underlying clouds separately," Does "with the surface" mean CRE is simply the difference in upwelling flux with and without TMLCs present (and no cloud underlying the TMLC)? Does "with underlying clouds" mean that underlying clouds impact only the overcast-sky upwelling flux (and the clear-sky upwelling flux is left unchanged)? Or is this a different CRE representing the difference in upwelling flux between a scene with only TMLCs and a scene with only low-level clouds? It is hard for the reader to understand the CRE calculations without more explicit and clear language.

Line 364: I still have a problem with this statement, which trivializes the model-observation difference and gives the impression that what you've done is all that is needed to do in order to reconcile models with observations. "...output, and have thus removed..." should be replaced with "...output. As a first step toward making a reasonable comparison, we have removed..." Better yet, since you are only showing 2 models anyway, use two models that actually implemented the Calipso simulator and perform the same screening.

Review of "The forgotten clouds of the tropical middle troposphere"

The authors have addressed most of my comments, but a few issues still remain, which I list below. After addressing these comments, I believe the paper will be suitable for publication.

We would like to thank the reviewer for the helpful comments that substantially improved the manuscript. We very much appreciated the detailed remarks and hope to have addressed all raised issues.

Lines 15-16: "have a cooling effect on climate that could be as large as cirrus (but of opposite sign)." Is awkward. Suggest "have a cooling effect on climate that could be as large in magnitude as the warming effect of cirrus"

Corrected as suggested.

Lines 155-158: "Although the albedo of low-level clouds is predicted to increase due to a changing concentration of cloud condensation nuclei, their albedo may decrease in the tropics due to the cloud adjustment to increasing CO₂." This statement needs to be supported with references. Also, the first part of the statement does not reflect the current thinking in this area. Low-level cloud albedo is expected to rise at mid- to high-latitudes owing primarily to warming-induced ice-to-liquid transitions, and secondarily to increases in the adiabatic water content of clouds as the warmer atmosphere becomes more moist. These are not dependent on changing CCN, which may or may not increase with increasing CO₂.

Since we focus our study on the tropics and it remains pretty unclear how the overall albedo of low-level clouds would respond in a warming climate, we removed the statement about a potential rising of the global albedo of low-level clouds. Therefore, the sentence is "Moreover, their albedo may decrease in the tropics due to the cloud adjustment to increasing CO₂". We added a reference to

support this statement.

Lines 210-211: I do not understand this statement: "first, we compute the radiative effect of TMLCs with the surface and underlying clouds separately," Does "with the surface" mean CRE is simply the difference in upwelling flux with and without TMLCs present (and no cloud underlying the TMLC)? Does "with underlying clouds" mean that underlying clouds impact only the overcast-sky upwelling flux (and the clear-sky upwelling flux is left unchanged)? Or is this a different CRE representing the difference in upwelling flux between a scene with only TMLCs and a scene with only low-level clouds? It is hard for the reader to understand the CRE calculations without more explicit and clear language.

We now mention that the cloud radiative forcing model calculates a difference of radiative fluxes between two scenes: with and without the cloud layer examined (here, TMLCs). We also better explain how the final radiative effect of TMLCs is calculated: "first, we compute the radiative effect of TMLCs for two scenes separately, over the surface and underlying clouds. Second, the two distinct radiative effects are multiplied with the probability to find TMLCs in clear-sky (62%) and with an underlying cloud layer (38%). Finally, the sum of these two radiative effects is multiplied with the overall probability to find TMLCs in the tropics (6%, cf. Table 1)."

Line 364: I still have a problem with this statement, which trivializes the model-observation difference and gives the impression that what you've done is all that is needed to do in order to reconcile models with observations. "...output, and have thus removed..." should be replaced with "...output. As a first step toward making a reasonable comparison, we have removed..." Better yet, since you are only showing 2 models anyway, use two models that actually implemented the Calipso simulator and perform the same screening.

Since it seems pretty clear for everyone (the reviewer, the authors and other researchers that

attended the presentation on TMLCs at conferences) that GCMs cannot reproduce TMLCs because 1) even a high-resolution model (RAMS) is not able to properly represent them and 2) these models lack explicit treatment of deep convection and associated detrainment processes, partly due to coarse horizontal and vertical resolution, we decided to remove this last figure that is – as highlighted by the reviewer – not comparing properly the modeling results and observations.